# Lying in wait: the resurgence of dengue virus after the Zika epidemic in Brazil

Anderson Fernandes Brito [1,10], Lais Ceschini Machado[2,10], Rachel J. Oidtman [3,10], Márcio Junio Lima Siconelli [4,10], Quan Minh Tran [3], Joseph R. Fauver [1], Rodrigo Dias de Oliveira Carvalho[2], Filipe Zimmer Dezordi [2], Mylena Ribeiro Pereira[5], Luiza Antunes de Castro-Jorge[4], Elaine Cristina Manini Minto[6], Luzia Márcia Romanholi Passos[6], Chaney C. Kalinich[1], Mary E. Petrone [1], Emma Allen[1], Guido Camargo España[3], Angkana T. Huang [7], Derek A. T. Cummings[7], Guy Baele[8], Rafael Freitas Oliveira Franca [5,11], Benedito Antônio Lopes da Fonseca[4,11], T. Alex Perkins [3,11], Gabriel Luz Wallau [2,11✉] & Nathan D. Grubaugh [1,9,11✉]

After the Zika virus (ZIKV) epidemic in the Americas in 2016, both Zika and dengue incidence declined to record lows in many countries in 2017–2018, but in 2019 dengue resurged in Brazil, causing ~2.1 million cases. In this study we use epidemiological, climatological and genomic data to investigate dengue dynamics in recent years in Brazil. First, we estimate dengue virus force of infection (FOI) and model mosquito-borne transmission suitability since the early 2000s. Our estimates reveal that DENV transmission was low in 2017–2018, despite conditions being suitable for viral spread. Our study also shows a marked decline in dengue susceptibility between 2002 and 2019, which could explain the synchronous decline of dengue in the country, partially as a result of protective immunity from prior ZIKV and/or DENV infections. Furthermore, we performed phylogeographic analyses using 69 newly sequenced genomes of dengue virus serotype 1 and 2 from Brazil, and found that the outbreaks in 2018–2019 were caused by local DENV lineages that persisted for 5–10 years, circulating cryptically before and after the Zika epidemic. We hypothesize that DENV lineages may circulate at low transmission levels for many years, until local conditions are suitable for higher transmission, when they cause major outbreaks.

[1] Department of Epidemiology of Microbial Diseases, Yale School of Public Health, Yale University, New Haven, CT, USA. [2] Department of Entomology, Aggeu Magalhaẽs Institute, Fiocruz, Recife, PE, Brazil. [3] Department of Biological Sciences and Eck Institute for Global Health, University of Notre Dame, Notre Dame, IN, USA. [4] Internal Medicine Department, Ribeirão Preto Medical School, University of São Paulo, Ribeirão Preto, SP, Brazil. [5] Department of Virology and Experimental Therapy, Aggeu Magalhaes Institute, Fiocruz, Recife, PE, Brazil. [6] Municipal Health Secretary from Ribeirão Preto, Ribeirão Preto, SP, Brazil. [7] Department of Biology and Emerging Pathogens Institute, University of Florida, Gainesville, FL, USA. [8] KU Leuven Department of Microbiology, Immunology and Transplantation, Rega Institute, Laboratory of Evolutionary and Computational Virology, Leuven, Belgium. [9] Department of Ecology and Evolutionary Biology, Yale University, New Haven, CT, USA. [10]These authors contributed equally: Anderson Fernandes Brito, Lais Ceschini Machado, Rachel J. Oidtman, Márcio Junio Lima Siconelli. [11]These authors jointly supervised this work: Rafael Freitas Oliveira Franca, Benedito Antônio Lopes da Fonseca, T. Alex Perkins, Gabriel Luz Wallau, Nathan D. Grubaugh. ✉email: gabriel.wallau@cpqam.fiocruz.br; nathan.grubaugh@yale.edu

Dengue virus (DENV) is a mosquito-borne virus in the family *Flaviviridae* that is classified into four serotypes (DENV serotypes 1–4 (DENV-1–4)). DENV is endemic to many of the tropical and subtropical regions of the world, putting ~50% of the global population (nearly 4 billion people) at risk of annual infections and leading to a growing number of outbreaks of dengue disease[1,2]. As the symptoms of dengue often overlap with many other diseases, the emergence of new mosquito-borne viruses within dengue-endemic regions can often be hard to detect. A perfect example is Zika virus (ZIKV), another flavivirus that circulated alongside DENV in Brazil for more than a year before apparent human infections reached a point where it was detected by surveillance systems[3,4]. As a consequence, during this period of "cryptic transmission" from late 2013 to 2015, ZIKV spread to over 40 countries in the Americas, reaching its peak epidemic in 2016[3,5–9]. Interestingly, the decline of Zika cases in Brazil during 2017 and 2018 was mirrored by a decline in reported dengue cases; and similar patterns were observed throughout the America[10]. The 2 years of abnormally low dengue incidence were followed by a synchronized resurgence of dengue cases in 2019 across the Americas. In that year, a record high 3.1 million dengue cases were reported throughout the region (2.1 million in Brazil alone), with an incidence of 321 cases per 100,000 habitants[11]. In 2019, dengue also surged worldwide, with major outbreaks also reported in Asia (e.g., Pakistan and Bangladesh) and Africa (e.g., Benin, Côte d'Ivoire, Senegal, and Tanzania)[12]. These fluctuations in dengue incidence raise several important questions: (1) Did DENV transmission really decline in Brazil during 2017–2018, or only the reported dengue cases? (2) If the decline in transmission was real, was it caused by population immunity imposed by previous dengue or Zika outbreaks? (3) Was the 2019 resurgence in dengue caused by new DENV introductions? Considering the global distribution of both ZIKV and DENV[2,13], answering these questions can help us to better prepare for and control future outbreaks.

Several anthropological, ecological, and immunological factors may explain the decline of dengue incidence following Zika outbreaks. Human interventions, such as enhanced mosquito control, or changes in human behavior in response to the Zika epidemic could explain the decline in reported dengue cases[14,15]. Ecological factors, such as local humidity and temperature, may also impact the transmission dynamics of arboviral diseases[16]. As the mosquito vectors for DENV, *Aedes aegypti* and *Ae. albopictus*, can transmit ZIKV and chikungunya virus (CHIKV, an unrelated alphavirus), changes in climate would alter the transmission of all three viruses[14–16], and could be at least partially responsible for the fluctuations in DENV. Finally, immunological cross-protection via flavivirus antibodies can also impact epidemics[17], and the potential effects of a previous exposure to ZIKV on secondary DENV infections have been debated. Previous studies suggest that pre-exposure to ZIKV or DENV could (1) lead to severe dengue disease due to antibody-dependent enhancement[18,19], as commonly observed in secondary dengue infections by distinct dengue serotypes[17,20]; and/or (2) provide partial protection, leading to less severe disease[15,21,22]. Since about one-fifth of DENV infections result in clinically apparent symptoms[2,23], high levels of cross-protection could lead to lower number of severe infections, and even higher underreporting. This last scenario could explain the declines in dengue cases observed in countries hard hit by DENV and ZIKV prior to dengue resurgence in 2019, like in Brazil[24], Bolivia[7], Suriname[6], and Nicaragua[5].

The resurgence of dengue raises questions about the origin of the 2019 outbreaks in Brazil. All four DENV serotypes have circulated throughout the Americas since their reintroductions into the western hemisphere in the 1970s–1980s[25–29]; however, in more recent years, DENV-1 and DENV-2 have become predominant in Brazil[30]. Thus, it is unclear if the recent outbreaks were caused by existing DENV lineages cryptically circulating in Brazil before and during the years of low reported dengue cases, or if they were the results of new DENV lineages introduced from other countries. Historical dengue outbreaks caused by DENV-2 in Puerto Rico, for example, have been caused by both reintroductions (clade replacement) and local survival through periods of low transmission[31]. The distinction is important for both our basic understanding of DENV maintenance and developing targeted control methods.

To answer these questions, we combined genomic, epidemiological, and ecological data to investigate the resurgence of dengue across Brazil, and within distinct geographic regions in the country. Our estimates of annual DENV force of infection (FOI) show that low population susceptibility as a result of historical DENV transmission could have contributed to the periods of low incidence in 2017–2018. We then sequenced DENV-1 and DENV-2 from 69 clinical samples collected during 2010, 2018, and 2019 from Northeast and Southeast Brazil to identify the origins and trace the spread of the viruses during the resurgence in 2019. We found that the recent dengue outbreaks were caused by DENV lineages circulating in Brazil before the Zika epidemic, representing lineages that endured the period of low transmission. Furthermore, the dispersal velocity of these persistent DENV lineages peaked in 2016 and slowed with the decline of Zika and dengue in 2017. Together, our results indicate that DENV established cryptic transmission for more than 5 years before causing outbreaks in 2019, perhaps due to the dynamics of population immunity or broad public health measures in the aftermath of the Zika epidemic[21,32], scenarios that may have occurred across the Americas.

## Results

**Patterns of dengue incidence in Brazil.** While the annual dengue incidence in Brazil varies by year, the general trend in the twenty-first century has been an increasing burden of disease (Fig. 1A). In 2016, dengue cases exceeded 1.6 million, which was a record at the time[33]. In 2017 and 2018, the number of annually reported dengue cases declined to ~250,000, the lowest since 2005 (~200,000; Fig. 1A). In 2019, dengue cases again rebounded, setting a new record of ~2.1 million reported cases in Brazil (Fig. 1A). We show that similar patterns of dengue incidence were also observed in each of the five geographic regions of the country (Fig. 1B, C).

Our investigation focuses mainly in two states: Paraíba, in Northeast Brazil, and São Paulo, in Southeast Brazil. Located in different regions of the country, distinct municipalities in these states experienced major dengue outbreaks in recent years, after years of low dengue incidence. Looking back in 2014, the number of reported dengue cases was low in both states, but in 2015 and 2016, dengue cases surged in many municipalities (Fig. 2A, B). In 2017, dengue incidence declined to record lows in Paraíba and São Paulo, reaching levels that resemble those of 2014 (Fig. S1). In 2018, some municipalities in both states experienced major dengue outbreaks, but it was in 2019 that dengue resurged at full strength, returning in record highs in São Paulo, while in Paraíba dengue resurged in specific municipalities, especially around the metropolitan area of the capital city, João Pessoa.

From 2016 to 2018, Zika incidence followed a similar pattern of surge and decline as dengue. After Zika was introduced in the country in late 2013[3,8], and spread undetected until May 2015[34], the number of Zika cases peaked in March 2016 (Figs. 1A and S2). The total number of Zika cases in Brazil and elsewhere during the pandemic is significantly underreported in part due to misdiagnosis as dengue from their overlapping symptoms and

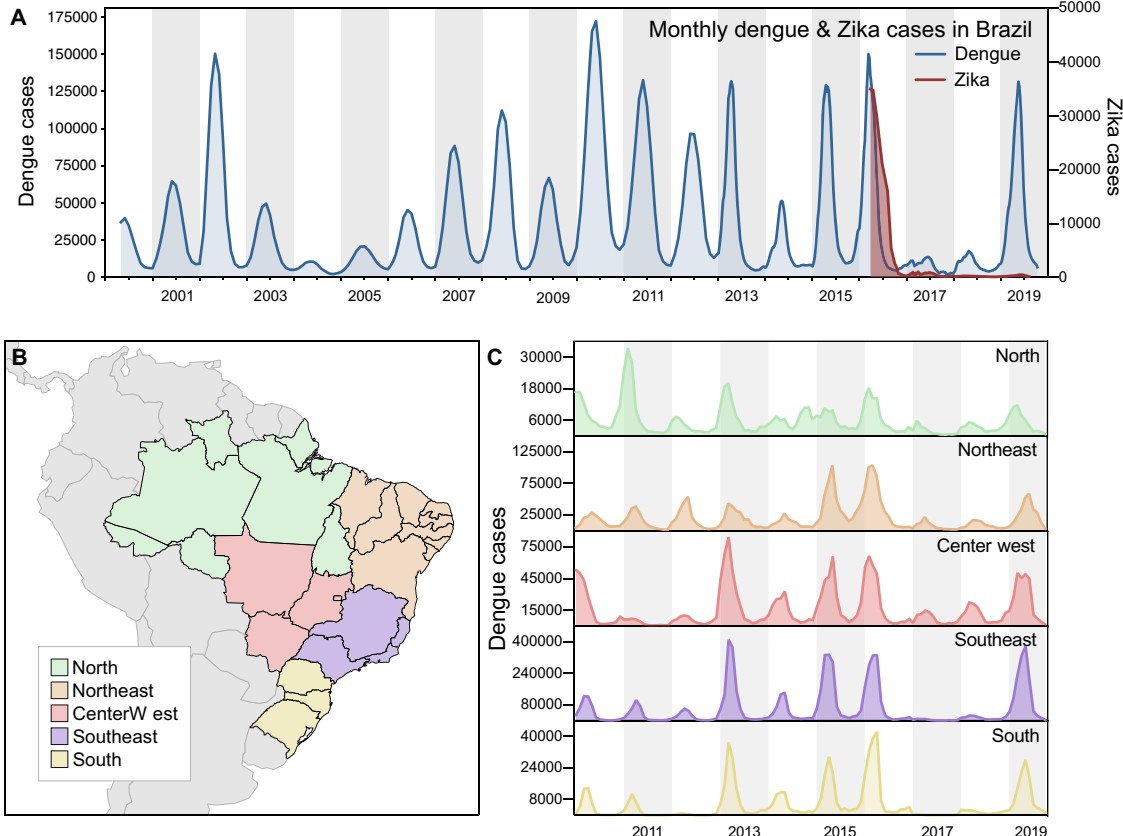

**Fig. 1 Dengue resurgence in Brazil. A** Monthly dengue and Zika cases in Brazil, from 2000 to 2019 (*y*-axes not in the same scale). **B** Map of geographic regions in Brazil, with arrows highlighting two states included in our study: Paraíba, in Northeast Brazil; and São Paulo, in Southeast Brazil. **C** Dengue cases per geographic region in Brazil, from 2010 to 2019 (*y*-axes not in the same scale).

cross-reactivity of many serological assays[35]. The synchrony of decrease of Zika and dengue cases following the peak in 2016 raises questions about the potential roles of flavivirus immunity, public health measures, and/or ecological factors that could explain these trends.

**Annual dengue virus force of infection**. The trends in dengue incidence could be caused by DENV transmission dynamics, vector control interventions, and/or biases in surveillance systems (Figs. 1 and 2). Commonly, only symptomatic infections are detected by surveillance, and the probability of experiencing dengue fever or severe dengue depends on whether an individual is experiencing a primary, secondary, or postsecondary DENV infection[36]. Changes in the demography and recent history of DENV transmission in a population can, through this mechanism, generate interannual variability in the probability that individuals experience dengue fever or severe dengue in a given year[37–39]. To determine whether natural variation in these factors could explain the decline in dengue incidence in 2017 and 2018, we estimated the FOI of DENV transmission on an annual basis from 2002 through 2019. This metric, defined as the rate at which susceptible individuals become infected, accounts for the aforementioned natural changes in the probability of experiencing dengue fever or severe dengue, and provides a more direct measure of DENV transmission than reported incidence.

Our estimates of FOI indicate that transmission was relatively low in 2017 and 2018, but not unprecedentedly so (Fig. 3). When we ranked years (between 2002 and 2019) by their FOI across the 100 replicates that comprised our estimates, we found that 2017 and 2018 were nearly always in the lower half of years in all five

regions, but the lowest only in the North region in 2018 (Fig. S3). In terms of magnitude, FOI was well above zero in the North, Northeast, and Center-West regions in those years (Fig. 3). In the Southeast and South regions, the magnitude of FOI was much lower, although that was not inconsistent with the very low magnitude of FOI in those regions in prior years (e.g., 2004–2006, 2008, 2009, and 2012 in the South region). However, susceptibility declined markedly in all regions over the period of our analysis (Fig. S4), meaning that an equivalent FOI would be expected to result in a smaller number of infections in 2017 and 2018 than in previous years with low FOI. Thus, after accounting for changes in susceptibility over time (Fig. S4), the probability that infections resulted in reported cases (Fig. S5), and serological data (Fig. S6), our estimates suggest that DENV transmission occurred in 2017 and 2018 at levels that were generally consistent with the lower bounds of expectations based on its recent history in Brazil.

**Mosquito-borne virus transmission potential**. If DENV infections prior to 2017 were enough to provide immunological protection against severe manifestations of dengue, then similar patterns of dengue decline should follow throughout the Americas where dengue incidence was previously high. We show that the drop in dengue incidence in 2017–2018 was also observed in many other regions in Latin America and the Caribbean, demonstrating this pattern was not unique to Brazil (Fig. 4A). However, there could be other factors contributing to the patterns of dengue incidence that are unrelated to population immunity or susceptibility, such as: (1) improved mosquito control and public health awareness campaigns implemented to reduce mosquito-

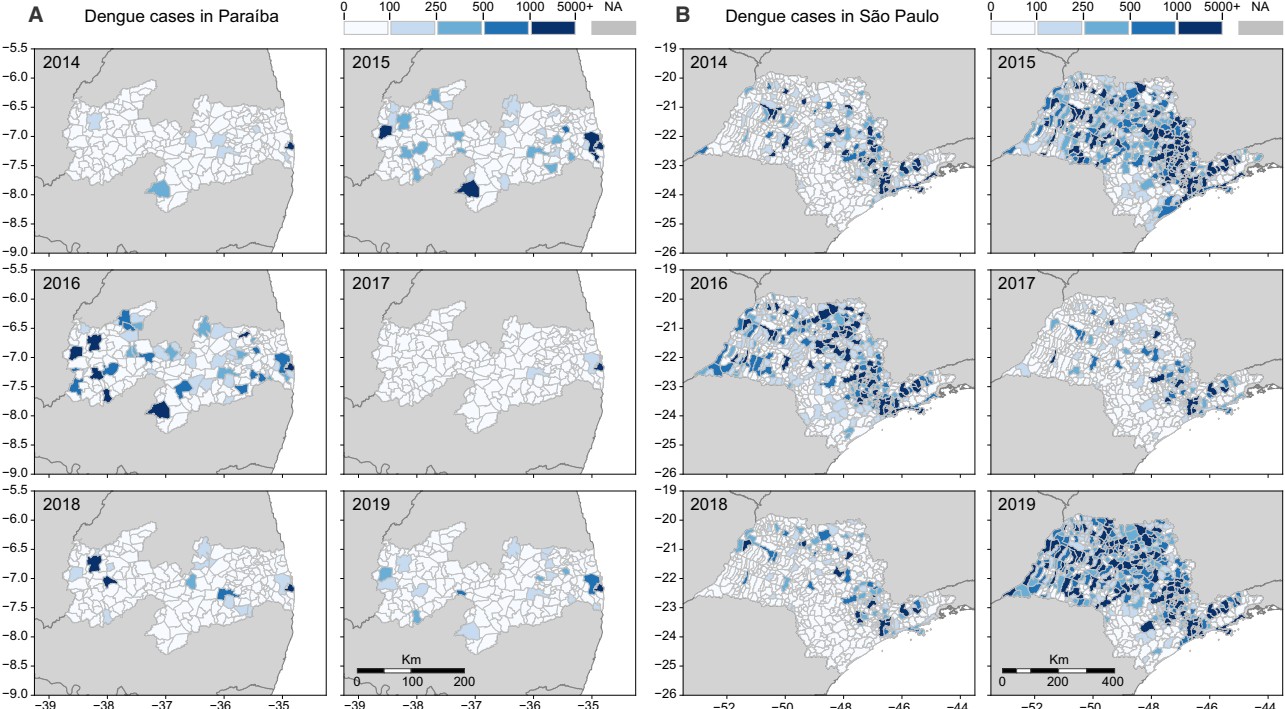

**Fig. 2 Reported dengue cases per municipality in Paraíba and São Paulo states, 2014–2019.** The dynamics of dengue in **A** Paraíba and **B** São Paulo between 2014 and 2019 followed similar cyclic patterns observed throughout Brazil: a decline in cases in 2014, followed by 2 years of high incidence (2015–2016), succeeded by 2 years of low incidence (2017–2018), with a resurgence of dengue in 2019. As in Figs. 5 and 6, the three main cities included in our study are highlighted in each state. In **A** Paraíba, we show: (1) João Pessoa; (2) Campina Grande; and (3) Coremas. In **B** São Paulo, we highlight: 1. Ribeirão Preto; 2. Araraquara; and 3. São José do Rio Preto. Normalized dengue incidence per 100,000 population is shown in Fig. S1.

borne virus transmission and (2) changing ecological factors that govern mosquito-borne transmission.

To evaluate the potential impacts of public health initiatives and/or human behaviors (e.g., treatment seeking) prior to and during the period of dengue decline, we investigated the patterns of reported cases of chikungunya (Fig. 4B). CHIKV is also transmitted by *Ae. aegypti* and *Ae. albopictus*, but as an alphavirus, it would not be affected by previous exposure to the flaviviruses, ZIKV and DENV. Thus, if CHIKV followed similar patterns as ZIKV and DENV, it would suggest that systematic changes—ecology, vector control, and/or surveillance—would have also played a large role. We found that chikungunya cases, which also peaked in Brazil during 2016, gradually declined through 2019, although not to the same degree as dengue in 2017 (Fig. 4B). While the gradual decline in chikungunya cases in Brazil could be related to a rising immunity to CHIKV, the holistic public health actions implemented to control the Zika threat in Brazil[40] could have also impacted the transmission of CHIKV and other mosquito-borne viruses.

As dengue outbreaks within a region can be synchronous, driven in part by large climatic shifts (e.g., El Niño)[41–44], we sought to determine if recent ecological changes could have altered virus transmission potential by *Ae. aegypti* mosquitoes. For this analysis we measured the Index P, a metric of transmission potential of *Ae. aegypti* mosquitoes, inferred using temperature and humidity data (Fig. 4C)[45]. We used Index P to estimate transmission in major urban areas in the Northeast and Southeast Brazil historically affected by dengue (Fig. 2), and from where we obtained clinical samples for sequencing in this study (Figs. 5 and 6). Our analysis shows that Index P was not notably different between 2016 and 2019 (Fig. 4C), and does not account for the differences in dengue cases in those regions (Fig. 1C). Thus, based on dengue incidence in the Americas, chikungunya

incidence in Brazil, and mosquito transmission potential, we conclude that the changes in dengue patterns were likely not due to climate, but could have been affected by broad control measures.

**Virus genomics reveals the timing of the DENV outbreak lineages.** When dengue incidence resurged throughout Brazil in 2019 (Fig. 1), it was unclear if it was caused by new DENV introductions or by lineages already established in the country, which survived 2 years of low transmission. To investigate, we sequenced DENV from the recent outbreaks in Northeast Brazil (mainly affected by DENV-1) and Southeast Brazil (mainly affected by DENV-2)[30], regions that typically have higher dengue case counts than the rest of the country (Fig. 1C). We sequenced 43 DENV-1 genomes collected from 2018 to 2019 dengue cases identified in the Northeast states of Paraíba and Alagoas. From the Southeastern state of São Paulo, we sequenced 3 DENV-1 genomes from 2018, 4 DENV-2 genomes from 2010, and 19 DENV-2 genomes from 2019 (Table S2). The DENV samples from São Paulo were sequenced by placing serum from dengue cases on FTA filter paper cards (which thereby inactivated the virus[46]) and shipping them to the USA (see "Methods"; Fig. S7), thereby highlighting how this simple method[47] can be used to supplement virus sequencing when local capacities are limited.

To uncover the origins of the recent dengue outbreaks, we combined our 46 DENV-1 and 23 DENV-2 genomes with others that were available in public databases (see "Methods"), and inferred time-resolved phylogenetic trees (DENV-1 = 200 total genomes, Fig. 5; DENV-2 = 220 total genomes, Fig. 6; root-to-tip estimates of evolutionary rate, Fig. S8). Our data show that the time to most recent common ancestor (TMRCA) for DENV-1 sequenced from Paraíba and Alagoas was mid-2012 (95% Bayesian credible interval (BCI) 2011.79–2013.33; Fig. 5), and

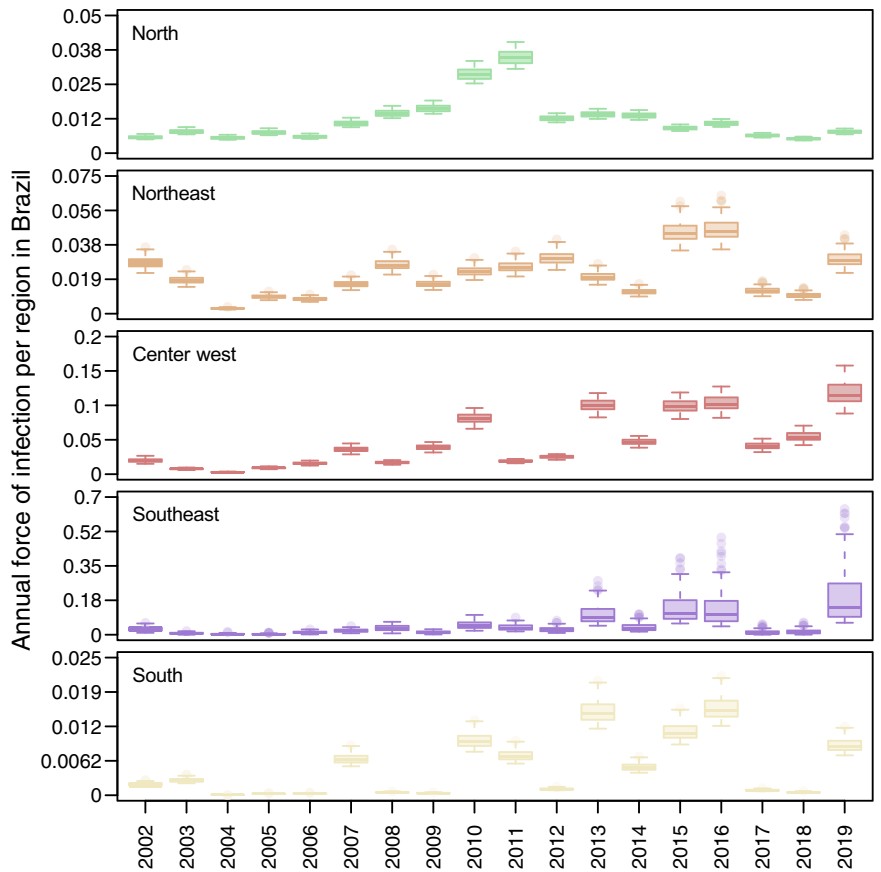

**Fig. 3 Dengue virus force of infection was low in 2017 and 2018 but consistent with other years in the recent past.** Values displayed here come from the set of 100 final estimates, which reflect the top 10% of 1000 parameter sets. These originally involved sampling $\vec{\gamma}_F$ and $\vec{\gamma}_S$ from assumed uniform priors (specified in Table S1) and then finding maximum-likelihood estimates of other model parameters. Boxes indicate the interquartile ranges and whiskers indicate the 95% quantiles of those 100 points, with dark lines representing median estimates. The regions and colors correspond to the map in Fig. 1B.

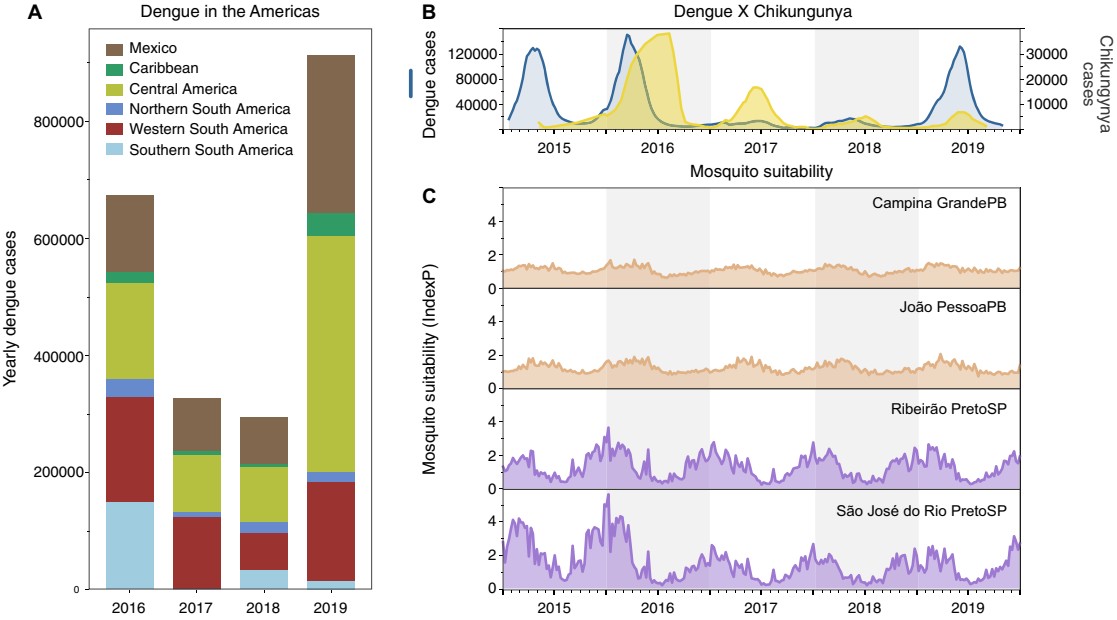

**Fig. 4 Low dengue cases in 2017–2018 not primarily due to surveillance or climate. A** Dengue cases from 2016 to 2019 reported to the Pan American Health Organization aggregated by region in the Americas. A sharp decrease in dengue cases was observed in 2017–2018 in distinct subcontinental regions in the Americas: Northern (from Venezuela to French Guyana), Western (from Colombia to Bolivia), and Southern South America (from Paraguay to Argentina), Caribbean, Central America, and Mexico. **B** In Brazil, the comparative epidemiological curves of dengue and chikungunya cases before and after major Zika outbreaks in 2016. **C** Measures of Index P, a mosquito-borne viral suitability index (transmission potential), estimated from distinct urban areas in Paraíba (PB, northeast Brazil) and São Paulo (SP, southeast Brazil) states.

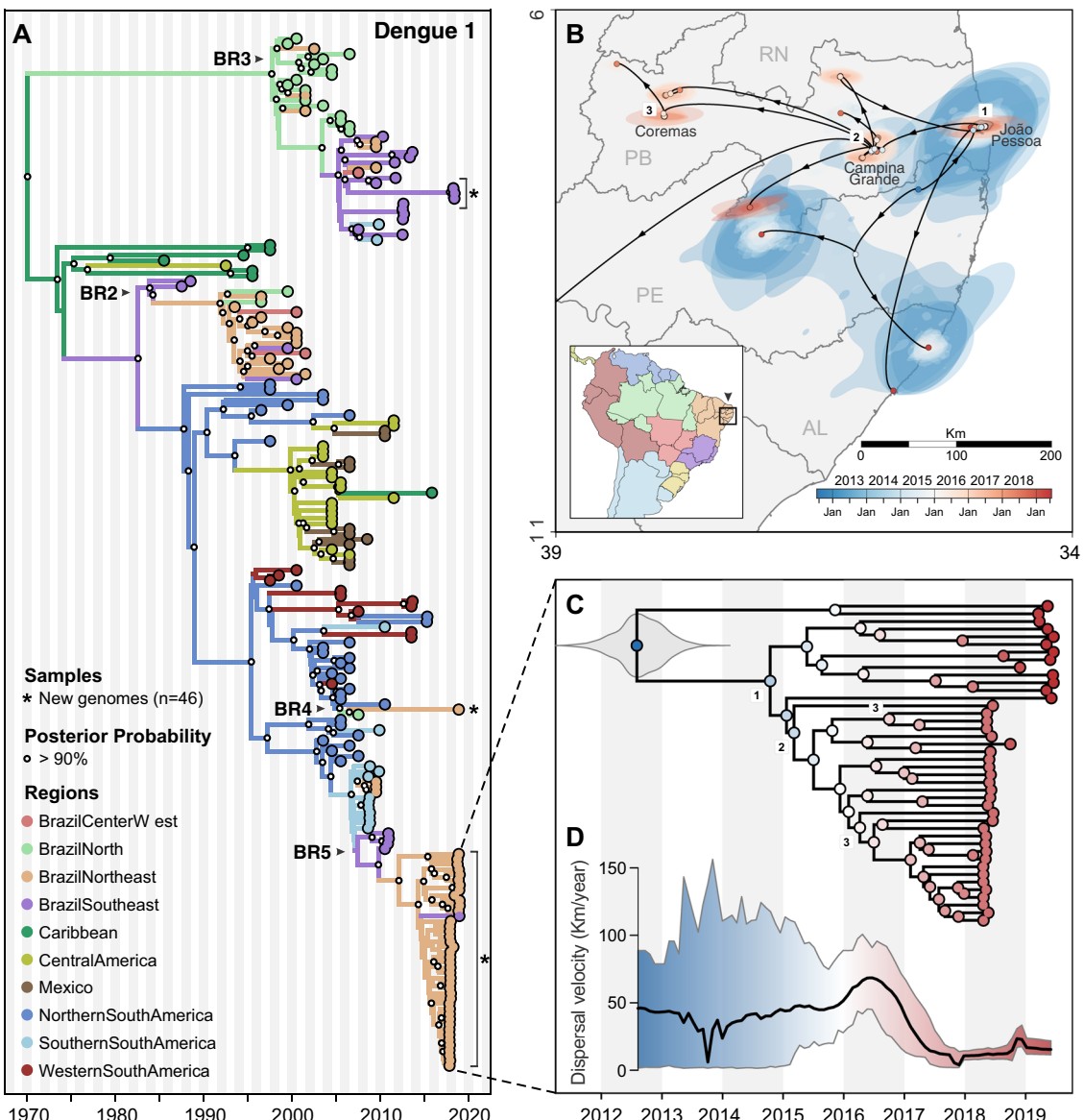

**Fig. 5 Regional emergence and cryptic transmission of DENV-1 causing recent outbreaks in Northeast Brazil. A** Time-resolved phylogeny of DENV-1 circulating in the Americas since 1986 (*n* = 200). Branch colors represent reconstructed ancestral locations using discrete phylogeography. BR2–BR5 represent lineages of DENV-1, numbered in sequential order based on their dates of introduction in Brazil, as previously described[49]. DENV-1 genomes sequenced in this study (*n* = 46) are highlighted with asterisks (*). **B** Continuous phylogeography showing the local spread of DENV-1 in the Northeast Brazil states of Paraíba (PB), Alagoas (AL), and likely Pernambuco (PE). Areas numbered as 1 (João Pessoa), 2 (Campina Grande), and 3 (Coremas) correspond to the main location where DENV-1 circulated in 2018–2019 (see Fig. 2A). Shaded areas represent uncertainties, expressed as the 80% highest posterior density (HPD) of the possible locations of origin of viral ancestors. **C** The main DENV-1 outbreak clade plotted as movement vectors in (**B**). The violin plot shows the posterior density interval for the TMRCA. Numbers refer to areas shown in (**B**). **D** Weighted lineage dispersal velocity through time, reaching its peak around June 2016, with mean velocity of 69.1 km/year (confidence interval, 45.07–90.91 km/year). To better depict the dynamics of spread in Northeast Brazil, the single vector leading to São Paulo was not considered in the calculations of dispersal velocity.

the TMRCA for DENV-2 sequenced from São Paulo was early 2014 (95% BCI 2013.33–2014.61; Fig. 6), which is independently confirmed[48]. As there are far more available DENV envelope protein coding sequences than whole genomes[49–51], we repeated our analysis using 1250 DENV-1 (Fig. S9) and 1202 DENV-2 (Fig. S10) envelope sequences reported in previous studies[49–51], and we found similar clustering patterns and ancestral phylogeographic origins. Overall, our data and analyses reveal a common pattern: lineages of DENV-1 and DENV-2 causing outbreaks in Southeast and Northeast Brazil were most likely introduced and descended from viruses circulating in those regions before the Zika epidemic.

**Origins and spatial dispersal of DENV-1 lineages circulating from 2012 to 2019.** Our phylogenetic analysis revealed that DENV-1 lineages causing the 2019 outbreak in Northeast Brazil had been circulating undetected in the region since at least mid-2012 (Fig. 5). Using near-complete genomes from DENV-1 isolates from Brazil and other countries in the Americas since the 1980s, we then explored (1) the genetic relatedness of DENV-1 genomes in the Northeast and Southeast, (2) how long the outbreak lineages have been circulating in Brazil, and (3) the patterns of within-region spread during the period of cryptic DENV-1 transmission.

DENV-1 was first detected in Brazil in 1981 (Fig. S9) and, up until the 1990s, it was the main serotype circulating in the

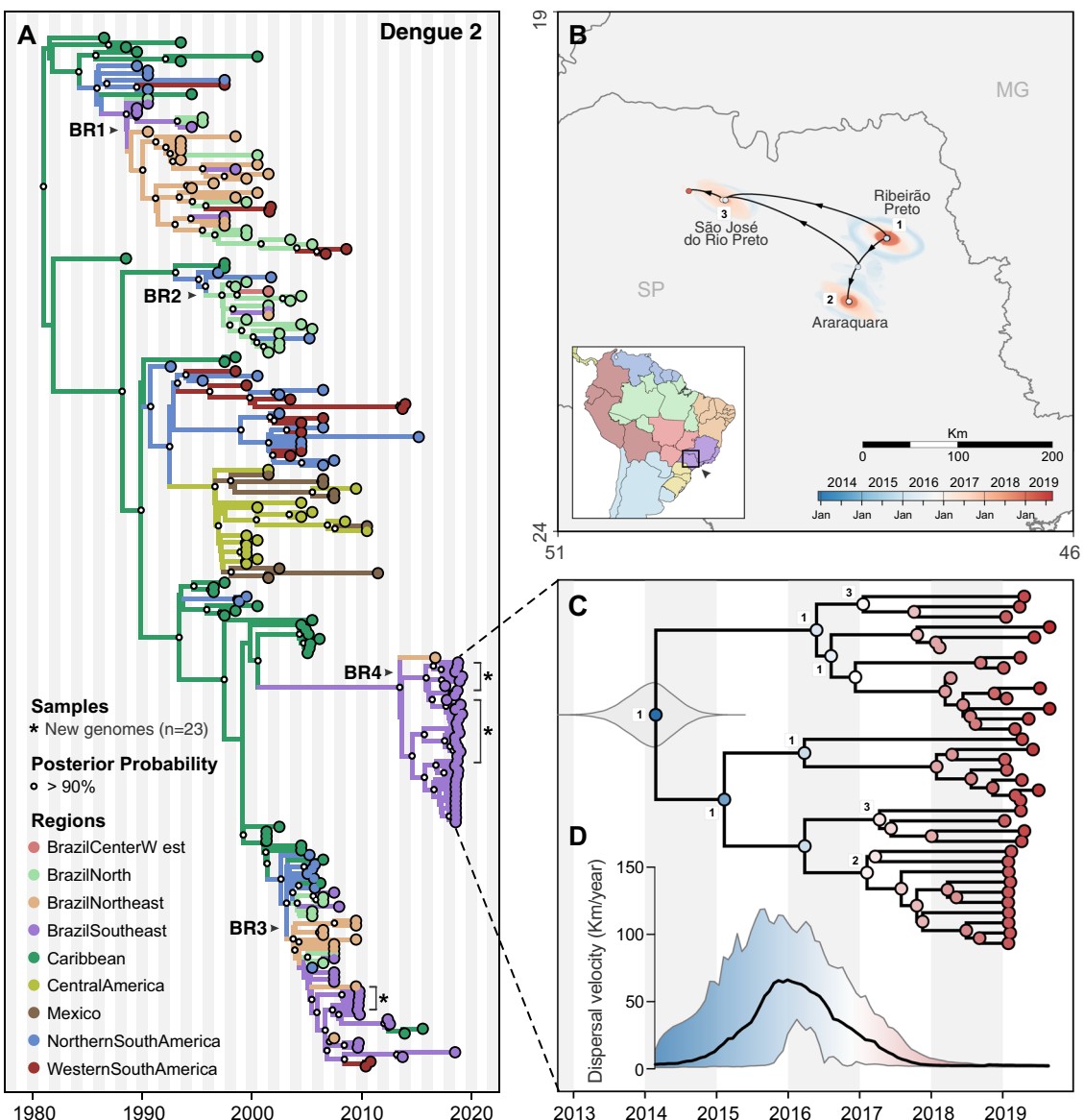

**Fig. 6 Regional emergence and cryptic transmission of dengue 2 viruses causing recent outbreaks in Southeast Brazil. A** Time-resolved phylogeny of DENV-2 circulating in the Americas since 1987 (*n* = 220). Branch colors represent reconstructed ancestral locations using discrete phylogeography. BR1–BR4 represent lineages of DENV-1, numbered in sequential order based on their dates of introduction in Brazil, as previously described[48,51]. DENV-2 genomes sequenced in this study (*n* = 23) are highlighted with asterisks (*). **B** Continuous phylogeography showing the local spread of DENV-2 in Southeast Brazil, state of São Paulo (SP). Areas numbered as 1 (Ribeirão Preto), 2 (Araraquara), and 3 (São José do Rio Preto) correspond to some of the main locations where DENV-2 circulated in 2018 (see Fig. 2B). Shaded areas represent uncertainties, expressed as the 80% highest posterior density (HPD) of the possible locations of origin of viral ancestors. **C** Main DENV-2 outbreak clade in (**A**), plotted as movement vectors in (**B**). The violin plot shows the posterior density interval for the TMRCA. Numbers refer to areas shown in (**B**). **D** Weighted lineage dispersal velocity through time, reaching its peak around January 2016, with mean velocity of 66.2 km/year (confidence interval, 31.3–103.5 km/year).

country along with DENV-2[52]. The main DENV-1 lineages sequenced from Brazil are classified into five clades: BR1–BR5, following their order of introduction[49,50] (Figs. 5A and S9). Our analysis shows that the 2019 DENV-1 outbreak in Northeast Brazil, and at least one case in Southeast Brazil, was primarily caused by viruses from lineage BR5, which have been circulating in Brazil for more than a decade (TMRCA = 2005–2007; Fig. 5A). Thus, the introduction of the BR5 lineage into Northeast Brazil around mid-2012 came likely from a domestic source. In addition, one of our sequenced DENV-1 genomes from Alagoas state (Northeast Brazil) clustered with the BR4 clade, and three DENV-1 genomes from São Paulo state (Southeast Brazil) clustered with the BR3 clade, indicating the continued co-

circulation of multiple DENV-1 lineages in Brazil (Figs. 5A and S9).

To reconstruct the regional dynamics of DENV-1 lineage BR5 since its introduction into Northeast Brazil, we performed continuous phylogeographic analysis to map viral spatiotemporal spread (Fig. 5B). We found that DENV-1 circulated undetected in the João Pessoa metropolitan area for at least 1 year through 2014 (region 1, in Figs. 2A and 5B), and then moved toward the metropolitan region of Campina Grande, another important urban area in Paraíba (region 2, in Figs. 2A and 5B). In this region, DENV-1 established local transmission chains in surrounding municipalities from 2015 to 2016 (Figs. 2A and 5B), and also spread back to Southeast Brazil (São Paulo state), where

it was detected in 2019 (Fig. 5A, in purple). In 2015, this lineage also spread toward the westernmost part of the state, where it circulated for at least 2 years before being detected by sequencing in 2018 (region 3 in Fig. 2A).

To estimate the dispersal velocity of cryptic spread, we used spatiotemporal information extracted from thousands of posterior trees inferred to investigate DENV-1 local spread (Fig. 5C, D). Starting in 2015, when DENV-1 dispersed toward the western part of Paraíba state (regions 2–3, Figs. 5B, C and 2A), the lineage BR5 progressed to reach its highest dispersion rate: ~69.1 km/year, in mid-2016 (Fig. 5D). It was followed by a large drop in dispersal velocity, when mainly local, short-range transmissions were predominant in the region after late 2017 (Fig. 5D). Thus, our analysis suggests that the spread of DENV-1 in the region declined after the major outbreaks in the state, in 2015 and 2016, and the outbreak in 2019 was likely mainly caused by locally established viruses. This observation is supported by both the case data (Fig. 2A) and by our phylogeographic analysis (Fig. 5B, C).

**Origins and spatial dispersal of DENV-2 lineages circulating from 2013 to 2019.** Similar to DENV-1 during the outbreak in Northeast Brazil (Fig. 5), we found that the 2019 dengue outbreak in the Southeast state of São Paulo was caused by a DENV-2 lineage established in the region since at least 2014 (Fig. 6). To investigate the spatial dynamics of DENV-2 in Southeast Brazil, we performed similar phylogeographic analyses as described above.

Combining our 23 DENV-2 genomes from Ribeirão Preto (region 1 in Figs. 2B and 6B) with 20 genomes recently sequenced by de Jesus et al.[48] from Araraquara (region 2) and São José do Rio Preto (region 3) municipalities in São Paulo state, we found that the 2019 outbreak was primarily caused by a different DENV-2 lineage (BR4) than the one detected during the 2010 outbreak in the same area (lineage BR3; Figs. 6A and S10). However, there is evidence that the DENV-2 BR3 clade, which emerged in Brazil around 2004, is still circulating in Southeast Brazil (Fig. 6A)[48]. The Caribbean is the likely origin of DENV-2 ancestors that now form the BR4 lineage, which was introduced into Brazil no later than 2014 (95% BCI, 2013.33–2014.61, Fig. 6C).

We reconstructed the regional dispersal history of DENV-2 lineage BR4 to uncover how the viruses spread in Southeast Brazil during mostly cryptic transmission (Fig. 6B, D). Following the same approach used to investigate DENV-1 diffusion through time, we found that DENV-2 was most likely introduced in the municipality of Ribeirão Preto (region 1) around 2014, where it circulated for 1–2 years (Fig. 6B, C). Through 2015 this lineage reached its fastest dispersal (66.2 km/year by early 2016) as it spread toward Araraquara, São José do Rio Preto (regions 2–3; Fig. 6C, D), and likely to many other areas in São Paulo state (Fig. 2B). Once established in these new areas, DENV-2 circulated locally via short range, sustained transmission chains, as revealed by its low dispersal velocity through 2018–2019 (Fig. 6D). Similar to DENV-1 lineage BR5 in Northeast Brazil, the patterns of DENV-2 lineage BR4 dispersal in Southeast Brazil (Fig. 1) highlight that the 2019 dengue outbreaks were caused by locally established viruses, not new introductions.

## Discussion
### Resurgence of dengue following a period of low transmission.
Following the major Zika epidemic in 2016, dengue incidence remained low throughout the Americas for 2 years (2017–2018), resurging in Brazil and elsewhere in 2019 (Figs. 1 and 4). The reasons for the drastic declines in dengue cases have been debated[14,15], but evidence to explain such phenomena are scarce.

In this study, we combined genomic, ecological, and epidemiological data to address this question. We found that DENV transmission decreased in 2017–2018 to levels not inconsistent with other recent years (Fig. 3). Although our analysis does not offer a full explanation for the levels of transmission observed in 2017 and 2018, our results indicate that changes in environmental conditions or surveillance gaps were unlikely to have driven this pattern (Fig. 4). In addition, our reconstruction of population susceptibility (Fig. S4) indicates that susceptibility was lower in 2017 and 2018 than in other recent years with low FOI (e.g., 2004–2006), which exacerbated the apparently low numbers of reported cases in 2017 and 2018. Using viral genomes collected in 2010, 2018, and 2019, we uncovered the origins of recent dengue outbreaks in Brazil, which were caused by endemic DENV lineages cryptically circulating in Brazil for 5–10 years before causing outbreaks in 2019 (Figs. 5 and 6). Our findings revealed that DENV can endure periods of low population susceptibility and resurge when conditions are suitable.

### Low population susceptibility contributed to low dengue incidence in 2017–2018.
Our annual estimates of DENV FOI in the five regions of Brazil suggest that the number of susceptible individuals from 2017 to 2018 dropped to some of the lowest levels since 2002 (Fig. 3). A major caveat of this analysis, however, is that we estimated FOI using all dengue case counts, regardless of serotype. Thus our analysis does not capture the individual transmission dynamics of the four DENV serotypes, which are known to co-circulate in Brazil at least since 2010[53]. Serotype dynamics, or the interplay between DENV and ZIKV, could be one of the primary explanations for interannual variability in FOI (Fig. 3) that we characterized statistically but did not explain mechanistically. In other settings, antigenic differences among serotypes have been shown to drive serotype dynamics and temporal patterns of transmission more broadly[54,55]. Still, transmission of all DENV serotypes was low during 2017–2018, as suggested by the low reported overall incidence.

The decrease in dengue incidence in 2017–2018 appears to have resulted in part from population immunity building (Fig. S4) during years of high dengue incidence in the preceding decade (Figs. 1 and 2). The time series of dengue cases in Brazil shows that major declines of dengue occur after a few years of large DENV transmission and reported cases (Figs. 2 and 3), a pattern also observed in other countries[56–58]. Investigations to better understand the periodicity at which dengue may decline due to population immunity would be advantageous. It has been hypothesized that ZIKV-induced immunity could have a positive impact, offering cross-protection against symptomatic DENV infections[15,21]. This scenario, however, seems to be unlikely to fully explain the decline of dengue, especially in light of recent findings showing that intermediate titers of ZIKV antibodies may increase the risk of severe dengue disease on secondary infections by DENV[19]. If the dynamics of flavivirus population immunity contributed to the decline and resurgence of dengue in Brazil, it is important to emphasize that this may not be the only factor and our findings may not be generalizable to other regions.

### The roles of climate factors and public health interventions on dengue dynamics.
An important finding of our study was that environmental conditions were likely not the primary cause of the fluctuations in dengue incidence in Brazil. DENV transmission is highly dependent on climate factors, such as humidity and temperature[45]. By using climate-dependent and climate-independent data, we estimated mosquito suitability in four major urban areas in the states of Paraíba (Northeast) and São Paulo (Southeast region; Fig. 4C), where our viral samples came

from, and we found that conditions remained suitable for mosquito-borne virus transmission. Thus, it is unlikely that climate had a notable effect on the observed dengue changes during this study period.

On the other hand, public health interventions following the Zika epidemic could have impacted DENV transmission[15]. After increases in the number of Guillain–Barré syndrome and microcephaly in northeastern Brazilian states in late 2015, the Pan American Health Organization (PAHO) and the World Health Organization (WHO) deployed experts of the Global Outbreak Alert and Response Network to Brazil in November 2015[40]. With more than ten countries and territories being affected by a virus likely causing microcephaly and neurological syndromes[40,59,60], in February 2016 the WHO declared Zika a Public Health Emergency of International Concern. As a result, Brazil's Ministry of Health allocated additional funds to implement measures of mosquito control, which included campaigns to raise awareness, providing information about self-protection against mosquito-borne viral infections[61], and actions to eliminate mosquito breeding sites in vulnerable communities[62]. Similar public health initiatives to curb the Zika epidemic were implemented throughout the Americas, aiming at monitoring, preventing, and responding to outbreaks, focusing on: building diagnostic capacity, expanding vector control networks, and encouraging behavior changes with direct participation of the communities in Latin America and the Caribbean[40]. In this way, not only in Brazil but in other countries, the preparedness in response to the Zika epidemic in 2016 may also account for low transmission of mosquito-borne viruses, including DENV and CHIKV, in the following years. Unfortunately, data about the mosquito control programs, surveillance strategies, and human behavior from our study areas are not available, and could not be incorporated in our analyses. However, it is plausible to hypothesize that the aforementioned actions in response to Zika had a systemic impact in the transmission of other mosquito-borne viruses (Figs. 1A and 3A, B).

### Endemic dengue viruses lineages responsible for recent outbreaks.
Despite the low dengue incidence in 2017–2018, we found that DENV lineages already circulating in Brazil for 5–10 years reemerged to cause outbreaks in the Southeast and Northeast regions in 2019. This demonstrates that (1) major dengue outbreaks are not always caused by recent virus introductions and (2) DENV can cryptically survive during periods of low transmission. The 2019 outbreaks caused by DENV-1 in Northeast and Southeast Brazil included at least three endemic lineages, which were named sequentially based on their date of introduction in Brazil[49,50]: BR3, BR4, and mainly BR5 (Fig. 5A). The 2019 outbreaks caused by DENV-2 in Southeast Brazil (São Paulo state) included the lineages BR3 and mainly BR4 (Fig. 6), as also reported in a recent study[48]. Our continuous phylogeographic analyses for DENV-1 and DENV-2 indicate that following introduction, lineages likely go through periods of widespread regional dispersal before becoming established in an urban area, leading to a local outbreak.

Although our DENV genome sampling in Northeast Brazil was spatially heterogeneous, representing well the burden of dengue in 2018–2019 (Fig. 2A), our sampling in Southeast Brazil was highly focused on two major urban areas (regions 1 and 3, in Figs. 2B and 6B). This certainly has introduced biases in the patterns of DENV spread depicted in our continuous phylogenetic analysis for São Paulo state (Fig. 6B–D), which captures only a small fraction of the dynamics of DENV spread in that area. Future studies investigating the genomic epidemiology of DENV, or any pathogen, should ideally sample genomes commensurate

with the disease incidence in times and locations under investigation (Fig. 2), to ensure more realistic reconstructions of the geographic patterns of spread.

### Future directions and implications.
Our study suggests a process in which the endemic cycles of dengue outbreaks are not always caused by introductions of antigenically divergent DENV lineages immediately resulting in wide-spread transmission, but instead are likely many years in the making. How DENV lineages can persist undetected through long periods of low transmission is uncertain and should be further investigated. Furthermore, the outcomes of the present study could not directly identify what caused the fall and rise of dengue, observed in several countries[5–7,24] but our data support a hypothesis pertaining to the role of prior DENV immunity from previous dengue outbreaks, and perhaps public health interventions in response to the 2016 Zika epidemic as important factors leading to low dengue transmission in 2017–2018. Further research and cohort studies are required to evaluate whether the levels of exposure to DENV in 2016 provided substantial protection against DENV infection or severe disease outcomes in the following years, and if previous exposure to ZIKV or other flaviviruses played any role. Furthermore, data about governmental mosquito control programs and public health campaigns should be made openly available to better evaluate the efficacy of these activities on controlling outbreaks. Finally, we advise for sustained genomic surveillance of DENV to detect cryptic lineages in an effort to plan for dengue resurgences. Collectively, our study revealed new insights into endemic DENV transmission and spread, which provides a basis for developing targeted control strategies.

## Methods

**Ethics**. Human sampling from Paraiba and Alagoas states used in this study was approved for the Research Ethic Committee (REC) from the Aggeu Magalhaes Institute (IAM) under the CAAE number 10117119.6.0000.5190. Human sampling from São Paulo state used in this study was approved by REC from Hospital das Clínicas de Ribeirão Preto process number 12603/2006 and also by DAEP/CSE-FMRP-USP project number 19/2007. DENV sequencing and analysis of a de-identified and limited data set at the Yale School of Public Health were determined to be exempt from human research determined by a limited Yale Institutional Review Board (IRB) review (IRB protocol ID: 2000025320).

**Epidemiological data**. Dengue, Zika, and chikungunya case data per epidemiological week in Brazil were obtained from the Ministry of Health of Brazil, from its SINAN database[63] and its epidemiological bulletins[64], and shared by the Infectious Disease Dynamics Group at University of Florida (UF-IDD). Dengue case data per year in other countries in the Americas were obtained from PAHO[11]. Links to all data sets can be found in "Data availability." Epidemiological data were plotted in maps using the Python packages Geopandas (version 0.7.0) and Shapely (version 1.7.0).

**Serological data**. We used age-stratified serological data collated from two modeling papers that also conducted a literature search of published serological studies for DENV in Brazil[65,66]. Specifically, we recorded the positive cases and the total samples by age group. The data for each study all had at least one age group, and the positive cases were defined by lab tests that should detect the existence of IgG antibody in blood samples, such as IgG ELISA, plaque reduction neutralization test, or haemagglutination inhibition tests. Overall, the collated data geographically cover all geographic regions except for the South. Studies ranged from 1991 to 2015, with additional details available in Table S3.

**Estimating force of infection**. We estimated the FOI, $\lambda_{y,r}$ of DENV on an annual basis for each year, $y$, between 2002 and 2019 for each of the five geographic regions in Brazil, $r$. These estimates were informed primarily by the number of cases of dengue fever, $F_{y,r,a}$, and severe dengue, $S_{y,r,a}$, for each year, region, and age, $a$. We calculated the contribution from $F_{y,r,a}$ and $S_{y,r,a}$ to the likelihood of a given series of annual forces of infection in region $r$, $\vec{\lambda}_r$, as:

$$L_{F,S}(\vec{\lambda}_r, \rho_{F,r}, \rho_{S,r}) = \prod_y \prod_a \frac{E\left[F_{y,r,a}|\vec{\lambda}_r, \varrho_{F,r}\right]^{F_{y,r,a}} E\left[S_{y,r,a}|\vec{\lambda}_r, \varrho_{S,r}\right]^{S_{y,r,a}} e^{-\left(E\left[F_{y,r,a}|\vec{\lambda}_r, \varrho_{F,r}\right] + E\left[S_{y,r,a}|\vec{\lambda}_r, \varrho_{S,r}\right]\right)}}{F_{y,r,a}! S_{y,r,a}!} \quad (1)$$

where $Q_{F,R}$ and $Q_{S,R}$ are region-specific probabilities of reporting a case of dengue

fever or severe dengue, respectively, and $E[F_{y,r,a}|\vec{\lambda}_r, \varrho_{F,r}]$ and $E[S_{y,r,a}|\vec{\lambda}_r, \varrho_{S,r}]$ are the expected numbers of cases of dengue fever and severe dengue predicted by the model. This assumes that observed cases of each type in each year, region, and age are independent Poisson random variables.

We calculated the expected number of cases of dengue fever as:

$$E[F_{y,r,a}|\vec{\lambda}_r, \varrho_{F,r}] = \left( \sigma^0_{y,r,a}(1 - e^{-4\lambda_{y,r}})\gamma^0_F + \sigma^1_{y,r,a}(1 - e^{-3\lambda_{y,r}})\gamma^1_F + \sigma^{2+}_{y,r,a}(1 - e^{-\lambda_{y,r}})\gamma^{2+}_F \right) \varrho_{F,r} N_{y,r,a} \quad (2)$$

where $\vec{\sigma}_{y,r,a} = \{\sigma^0_{y,r,a}, \sigma^1_{y,r,a}, \sigma^{2+}_{y,r,a}\}$ are probabilities of having experienced 0, 1, or 2 + previous DENV infections, $\vec{\gamma}_F = \{\gamma^0_F, \gamma^1_F, \gamma^{2+}_F\}$ are probabilities of experiencing dengue fever conditional on infection given 0, 1, or 2+ previous DENV infections, and $N_{y,r,a}$ is the population of individuals of age $a$ in region $r$ in year $y$. The calculation for $E[S_{y,r,a}|\vec{\lambda}_r, \varrho_{S,r}]$ is similar. We note that this assumes an even mixture of serotypes every year, which is a limitation of our analysis given that one or two serotypes are typically dominant at any given time.

Probabilities of having experienced a given number of previous DENV infections depend on the annual FOI experienced in each year of a person's life. In the first 2 years of life, $\vec{\sigma}_{y,r,0} = \{1, 0, 0\}$ and $\vec{\sigma}_{y,r,1} = \{e^{-4\lambda_{y-1,r}}, 1 - e^{-4\lambda_{y-1,r}}, 0\}$, because we assume that, at most, one DENV infection is possible in each year of life. For all other ages:

$$\sigma^0_{y,r,a} = \prod_{\zeta=1}^{a} e^{-4\lambda_{y-\zeta,r}} \quad (3a)$$

$$\sigma^1_{y,r,a} = \sum_{\xi=1}^{a} \left( \left( \prod_{\zeta;\zeta>\xi} e^{-4\lambda_{y-\zeta,r}} \right) \left( 1 - e^{-4\lambda_{y-\xi,r}} \right) \left( \prod_{\zeta;\zeta>\xi} e^{-3\lambda_{y-\zeta-1,r}} \right) \right) \quad (3b)$$

$$\sigma^{2+}_{y,r,a} = 1 - \sigma^0_{y,r,a} - \sigma^1_{y,r,a} \quad (3c)$$

where $\zeta$ is an index for a year in which an individual was not infected and $\xi$ is an index for a year in which an individual was infected. In addition to lifelong immunity to each serotype that an individual was infected by, this formulation accounts for cross-protection against all serotypes for 1 year after a person is infected. Knowing that the transmission of DENV in Brazil only resumed in 1980 after a long hiatus, the FOI before this time was set to 0[67]. Given that this time frame preceded the period of our analysis, FOI was set to a constant annual value for all years between 1980 and 2001, which we estimated as a single parameter for each region included in $\vec{\lambda}_r$. To avoid keeping track of all possible pairs of years in which an individual could have experienced two infections, we combined the proportion of individuals who had experienced two or more infections in Eq. (3c). Given that most infections resulting in dengue fever or severe dengue likely result from primary or secondary infections[23], we viewed this approximation as acceptable. Also in the interest of model tractability, we assumed that serotypes were evenly distributed within and across years and that they were identical in terms of infectiousness, virulence, and other traits, as have previous analyses of age-stratified case data[68].

In addition to $F_{y,r,a}$ and $S_{y,r,a}$, the likelihood of the parameters was informed by serological data. Whereas $F_{y,r,a}$ and $S_{y,r,a}$ provide information about relative patterns with respect to age and interannually, serological data provides unique information about the magnitude of infection. The contribution to the likelihood from each group $i$ in the serological studies ranging in age from age$_{l,i}$ to age$_{u,i}$ was:

$$L_{sero}(\vec{\lambda}_r) = \prod_i \frac{N_i}{P_i} p_i^{P_i} (1 - p_i)^{N_i - P_i} \quad (4)$$

where $N_i$ is the number tested, $p_i$ is the number seropositive, and $i$ is an index over all age groups in all studies from region $r$. The probability of being seropositive, $p_i$, was defined as the expected proportion seropositive, which equals:

$$p_i = \sum_{a=age_{l,i}}^{age_{u,i}} \left( 1 - e^{-4\sum_{y=t-a}^{t} \lambda_y} \right) Pop_{t,a}, \quad (5)$$

where $Pop_{t,a}$ is the proportion of people in study year $t$ at age $a$ within the age range from age$_{l,i}$ to age$_{u,i}$. Because serological data are a direct reflection of infection, they did not directly inform estimates of $Q_{F,R}$ and $Q_{S,R}$. The serological data did inform estimates of these parameters indirectly, however, via the information they provided about $\vec{\lambda}_r$.

We combined the likelihoods of the two data types into an overall likelihood according to:

$$L_{total} = L_{F,S}(\vec{\lambda}_r, \varrho_{F,r}, \varrho_{S,r}) L_{sero}(\vec{\lambda}_r)^{100} \quad (6)$$

which more heavily weighted the serological likelihood. We made this choice due to the fact that preliminary results indicated that the model fit poorly to the serological data without this adjustment. This was likely a result of the much larger number of data points on $F_{y,r,a}$ and $S_{y,r,a}$. Under this formulation of the likelihood, we used maximum-likelihood (ML) inference to estimate $\vec{\lambda}_r$, $Q_{F,R}$ $Q_{F,R}$, and $Q_{S,R}$ separately for each region $r$ conditional on 1000 sets of values of $\vec{\gamma}_F$ and $\vec{\gamma}_S$ drawn from uniform distributions specified in Table S1. For the South region, since there was no serological data there, we used estimates of $Q_{F,R}$ and $Q_{S,R}$ from the other four regions to inform its reporting probabilities through beta distributed priors fitted to estimates from the other four regions. We initialized the optimization algorithm by randomly sampling annual FOI values in $\vec{\lambda}_r$ from 0 to 0.2 and reporting probability values $Q_{S,R}$ and $Q_{S,R}$ from 0 to 1. To account for the fact that

some values of $\vec{\gamma}_F$ and $\vec{\gamma}_S$ produced better fits to the data than others, we focused our analysis on the parameter sets with the top 10% highest likelihood values.

**Mosquito transmission potential estimates.** We used the Bayesian approach developed by Obolski et al.[45] to estimate the weekly transmission potential (Index P). Briefly, we obtained daily temperature and relative humidity records for four cities in Brazil from openweathermap.org. We combined these data with ecological and entomological priors documented in the literature (Table S4) using the R package MVSE[45] to estimate daily transmission potential, which we then aggregated by week.

**Clinical sample collection and virus diagnostics**
*Overview.* Serum samples from dengue cases were collected from two regions in Brazil: São Paulo state in the Southeast, and Paraíba and Alagoas states in the Northeast. Sample collection location and dates are provided in Table S2. All patients presented at least with fever and two or more symptoms such as nausea, vomiting, rash, myalgia, headache, retroorbital pain, petechiae or positive tourniquet test, and leukopenia.

*São Paulo state.* Dengue patient serum samples were collected at Hospital das Clínicas from Ribeirão Preto Medical School and at Basic Health Units (UBDS) distributed throughout the municipality. DENV infection was first tested by a rapid IgM/IgG test (Dengue DUO ECO test, ECO Diagnóstica, Corinto, MG, Brazil) or NS1 ELISA (FOCUS Diagnostics, Cypress, CA, USA). Positive IgM/IgG or NS1 tests were then confirmed by RT-qPCR using the following protocol. RNA from the serum samples were extracted using KASVI (Mobius Life Science, Pinhais, PR, Brazil) according to the manufacturer's recommendations. RT-qPCR was performed in one-step using TaqMan Fast Virus 1-Step Master Mix (Applied Biosystems), following the manufacturer's recommendations. Primers and probes used were: FD1: 5′-GCACCAGGTGGGAAACGA-3′; RD1: 5′-TAGGAGCTTGAGG TGTTATGG-3′, DENV-1 probe: FAM-CTACAGAACATGGAACAAT-MGB and FD2: 5′-CTAAATGAAGAGCAGGACAAAAGGT-3′; RD2: 5′-ATCCATTTCCC-CATCCTCTGT-3′, DENV-2 probe: FAM-TGCAAACACTCCATGGTA-MGB. The reaction was performed in the 7500 Fast Real-Time PCR System (Applied Biosystems).

*Paraíba and Alagoas states.* Dengue patient serum samples were collected by the Public Health Central Laboratory (LACEN) of each state and sent to the Arboviroses Reference Laboratory at the Aggeu Magalhães Institute—Fiocruz in Recife, Pernambuco, for viral identification. DENV infections were detected by using the QIAamp Viral RNA kit (Qiagen, Hilden—Germany) to extract RNA from the serum samples followed by using a DENV-1-4 Real-Time RT-PCR Assay following the CDC's recommendations[69].

**Virus genome sequencing**
*Overview.* Due to limited next-generation sequencing (NGS) capacity available to the researchers in São Paulo state at the time of conducting this study, dengue clinical samples from this state were preserved on FTA filter paper cards and shipped to Yale University for metagenomic virus sequencing using an Illumina NovaSeq platform, as described below. From DENV samples collected in Paraíba and Alagoas states, a local Illumina MiSeq available at the IAM was used for amplicon-based virus sequencing.

**Metagenomic virus sequencing from samples stored on FTA cards**
*São Paulo state.* Frozen serum from DENV-1 and DENV-2 positive patients was thawed and 50 uL was mixed with 10–15 μL of RNA later (Ambion, USA). Around 55–65 μL of the mixture was spotted onto FTA Cards (Whatman™, GE Healthcare Bio-Sciences Corp, Piscataway, NJ, USA) and dried at room temperature. Each card (8.5 × 7.5 cm) was loaded with 16 different samples, respecting a minimal distance among the samples avoiding the sample cross-contamination. The cards were stored at room temperature until shipment to Yale University.

Upon receiving, FTA cards containing serum were frozen at −80 °C until processing. A scalpel was used to cut out and dice entire serum spots from FTA cards. The scalpel was sterilized between samples by submerging into 70% EtOH followed by submerging into 10% (v/v) bleach three times. To elute serum from FTA cards, diced pieces from each sample were placed in 350 μL of 1x TE, and rocked at 4 °C for 16 h. RNA was extracted from 200 μL of 1x TE elution using the Mag-Bind Viral DNA/RNA kit (Omega Bio-tek, USA) and eluted in 25 μL of water. DENV-2 samples were then screened via an RT-qPCR assay using FD2/RD2 primers with the iScript™ One-Step RT-PCR Kit With SYBR Green (Bio-Rad, USA). DENV-2 samples that produced a cycle threshold value > 33 and a unique melting curve were processed for NGS. Due to the smaller overall number of DENV-1 samples available ($N = 7$), all were processed for NGS.

DENV-1 and DENV-2 RNA eluted from FTA cards from São Paulo state were sequenced using an unbiased metagenomics approach on the Illumina NovaSeq at the Yale Center for Genome Analysis. To increase the overall quantity of input, we proceeded with a primer-extension pre-amplification SISPA approach to generate randomly amplified cDNA as described previously[70,71]. In detail, RNA was incubated with 1 μL Primer A (40 pmol/μL) and 0.5 μl of a 10 mM dNTP mix from

the SuperScript IV First-Strand Synthesis System (Thermo Fisher Scientific, USA) at 65 °C for 5 min. First-strand cDNA synthesis was carried out with this mixture using the SuperScript IV First-Strand Synthesis System without the addition of random hexamers according to manufacturer's protocol. Second-strand synthesis was performed using the Sequenase Version 2.0 DNA Polymerase System (Thermo Fisher Scientific, USA). A volume of 0.15 µl of Sequenase enzyme was incubated with 1 µl of 5X Sequenase Buffer, 3.85 µl of nuclease-free water, and 10 µl of the first-strand synthesis reaction at 37 °C for 8 min. An additional 0.15 µl of enzyme and 0.45 µl of Sequenase dilution buffer were added, and the total reaction was incubated for an additional 8 min at 37 °C. Samples were held at 4 °C until cleanup[70]. Second-strand synthesis reactions were cleaned using 0.8:1 ratio of Mag-Bind TotalPure NGS (Omega Bio-Tek, USA) beads to volume of sample. Following two washes of beads with 70% EtOH, cDNA was eluted into nuclease water. cDNA was incubated with 2.5 µL of primer B (40 pm/µL) and amplified using Q5 HS High-Fidelity 2x MM (New England Biolabs, USA) for 15 cycles according to manufacturer's instruction. PCR product was purified using a 0.8:1 ratio of Mag-Bind TotalPure NGS beads to volume of PCR mix. PCR products for each sample were quantified using the Qubit 1X dsDNA HS Assay Kit (Thermo Fisher, USA).

A total of 5 ng of dsDNA was used as input for library preparation using the KAPA Hyper Prep Kit (Roche, Switzerland) with NEXTflex™ Dual-Indexed DNA Barcodes. Following ligation of adapters and barcodes, individual samples were amplified using universal Illumina primers and KAPA HiFi DNA Polymerase (Roche, Switzerland) according to manufacturer's instructions. Libraries were purified using 1:1 ratio of Mag-Bind TotalPure NGS beads to volume of library, quantified using the Qubit 1X dsDNA HS Assay Kit, and pooled by equal concentrations. The final pool of libraries was size selected to 500–600 bp using the Pippin Prep (Sage Science, USA), and size distributions were confirmed using the Agilent Bioanalyzer High Sensitivity DNA Kit (Agilent Technologies, USA). Libraries were sequenced using a portion of a line on a NovaSeq at the Yale Center for Genome Analysis.

Elution and RNA extraction were attempted from a total of 96 serum samples stored on FTA cards. Of these, a total of 60 samples met the criteria for NGS (DENV-1 $N = 7$, DENV-2 $N = 53$). A total of 56 million read pairs ($2 \times 150$) were generated from 65 libraries, including five control libraries that were introduced at the sample elution stage and library preparation stage (Table S2). The proportion of reads aligning to each respective DENV reference genome ranged from 0 to 29.9%, although the majority of samples ($N = 52$) had <1% of reads passing QC aligned to the DENV reference genome. Each of the five control libraries had <15 total reads aligning to either the DENV-1 or DENV-2 reference genome. Of the 60 samples sequenced, a total of 25 (DENV-1 $N = 4$, DENV-2 $N = 21$) genomes meet our threshold of 70% at >10X coverage to be included in our analysis (Fig. S7).

**Amplicon-based virus sequencing**

*Paraíba and Alagoas states.* Total RNA from Northeast Brazil samples were sequenced using a whole-genome tiled PCR amplicon-based approach[72,73]. In brief, cDNA was synthesized using random hexamers (Invitrogen) and ProtoScript II Reverse Transcriptase (New England Biolabs). Then, the resulting cDNA was submitted to a multiplex PCR using primers covering the entire DENV-1 genomes designed using PrimalScheme (https://primalscheme.com/)[72]. DENV-1 primer sequences can be found in Table S5. PCR reactions were performed using four separate pools to minimize primer competition, using Q5 Hot Start High-Fidelity DNA Polymerase (New England Biolabs). The resulting ~400 bp amplicons were quantified using Qubit dsDNA HS Assay Kit (Thermo Fisher Scientific Inc.), the four separate PCR reactions were pooled, and 2 ng were processed through the Nextera XT Library Prep Kit (Illumina, San Diego, CA, USA). This material was sequenced on the MiSeq Illumina platform from the IAM—Fiocruz, employing a MiSeq reagent kit V3 of 150 cycles with a paired-end strategy.

**Sequencing data processing**. The goal of our bioinformatics pipeline was to generate complete or nearly complete DENV genome sequences for phylogenetic analysis using a reference-based consensus generation. Analysis was conducted on the high-performance computing clusters hosted by the Yale Center for Research Computing. Sequencing adapters, primer sequences (for amplicon-based sequencing), and low-quality reads were trimmed using Trimmomatic (version 0.39)[74]. Following quality control, paired reads were aligned to a reference DENV-1 (Accession Number: JX669465.1) and DENV-2 (Accession Number: KP188569.1) genome using bwa (version 0.7.17)[75]. The aligned reads were filtered and output as a.bam file using SAMtools (version 1.9)[76]. Sorted.bam files containing only mapped, paired reads were visualized in Geneious Prime (version 2019.0.3)[77]. Consensus sequences were generated in Geneious Prime and vcf-annotate (parameters --filter Qual = 20/MinDP = 100/SnpGap = 20) and vcf-consensus, where a minimum of ten reads were required to call a base at any given position in the genome and the threshold to insert ambiguous nucleotides into the consensus sequence was 75%. Regions of the genomes that did not reach the 10X depth of coverage cut-off had an "N" inserted into the consensus sequence. Final sequences that had >70% of the genome with >10X depth of coverage all showed to fit the expected molecular clock for viruses of their serotypes (Fig. S8), and were used for phylogenetic analyses.

**Virus genomic data selection and alignment**. To build our initial data sets, we downloaded from Genbank all complete genomes of genotypes of DENV (serotypes 1 and 2) commonly circulating in the Americas: DENV-1 genotype V ($n = 458$), and DENV-2 genotype AA ($n = 814$) complete genomes. Genomes without collection dates and location information were excluded. The sequences were aligned using MAFFT v7.471[78], manually curated and had their untranslated regions trimmed. Preliminary ML analyses were performed using IQTree v.1.6.12[79], with automatic model selection by the software. Using TempEst v.1.5.3[80], we inspected these genomes to identify major molecular clock outliers, which were removed from all downstream analyses. This yielded two high-quality data sets with genomes evenly sampled through time, allowing optimal clock signal (Fig. S7): DENV-1 ($n = 458$ genomes), and DENV-2 ($n = 700$ genomes). Based on these samples, smaller data sets were obtained by subsampling overrepresented clades, keeping only two representatives per deme, and adding our newly sequenced genomes.

**Discrete phylogeographic analyses**. To infer the evolutionary history of DENV-1 and 2, and understand the recent major outbreaks of dengue in Brazil, we performed Bayesian phylogenetic inference using BEAST v.1.10.4[81]. We performed model selection of distinct molecular clock models, and concluded that a relaxed clock model fits our DENV-1 and DENV-2 data better than a strict clock (Table S6). In this way, we specified a Bayesian skygrid coalescent tree prior with 50 grid points[82], with a GTR + Γ4 model of nucleotide substitution, and a relaxed molecular clock. Geographic data of sample origins were arranged in a geographic scheme of subnational (the five regions in Brazil) and subcontinental areas, such as Northern (from Venezuela to French Guyana), Western (from Colombia to Bolivia), and Southern South America (from Paraguay to Chile/Argentina), Caribbean, Central America, and Mexico. The ancestral origins of viruses circulating in the Americas were inferred using a reversible discrete phylogeography model[83]. Using the BEAGLE library (v3.1.0) to accelerate computation[84], Markov chain Monte Carlo (MCMC) sampling was performed for 100 million iterations, and convergence of parameters (ESS > 200) was assessed using Tracer v1.7[85]. After removing 10% burn-in, maximum clade credibility trees were summarized using TreeAnnotator v1.10.4, visualized using FigTree v1.4.4[86], and plotted using baltic v.0.1.0[87].

**Continuous phylogeographic analyses**. To infer the dispersal history of DENV lineages circulating in Northeast and Southeast Brazil in recent years, we used a continuous phylogeographic method[88] implemented in BEAST v.1.10.4[81]. These analyses focused on two monophyletic outbreak clades of DENV-1 and 2 genomes sequenced in this study (Figs. 5C and 6C). We used the same evolutionary model as described above, using the inferred heights (95% HPD) from such analyses (Figs. 5A and 6A) as normal prior distributions to set the TMRCA of the outbreak clades. Taking advantage of geographic coordinates associated with the outbreak viruses, a Cauchy relaxed random walk diffusion model was employed to infer coordinates associated with ancestors of viruses causing the recent outbreaks (internal nodes). MCMC sampling was performed for 150 million iterations, using the BEAGLE library (v3.1.0) to accelerate computation[84], and convergence was inspected using Tracer v1.7[85]. As a result, we obtained 10,000 trees with ancestral coordinates annotated at each node. After discarding 10% of the sampled trees as burn-in, the data recorded in 1000 trees were extracted and plotted in geographic space using the "seraphim" R package (retrieval date: November 18, 2019)[89].

**Reporting summary**. Further information on research design is available in the Nature Research Reporting Summary linked to this article.

## Data availability

The genomes generated in this study are available on NCBI (Accession Numbers: MT862854–MT862895; MW208040–MW208066), and listed in Table S2. Epidemiological data were downloaded from Brazil's Ministry of Health (http://www2.datasus.gov.br/DATASUS/index.php?area=0203&id=29878153), PAHO website, and from the GitHub account of the Infectious Disease Dynamics Group at University of Florida (UF-IDD, https://github.com/UF-IDD/dengue-Zika-chik_Americas). All relevant data used in Figs. 1, 2, and 4–6 are available on the following GitHub repository: https://github.com/grubaughlab/DENV-genomics/tree/master/paper1.

## Code availability

Codes used for processing raw sequences and generating consensus genomes are available on the following GitHub repository: https://github.com/grubaughlab/DENV-genomics/tree/master/paper1.

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

## Acknowledgements

We acknowledge members of the Grubaugh Lab, S. Taylor, and P. Jack for their ongoing discussions about this manuscript. This work was funded by the Hetch Award provided by the Yale Institute for Global Health (NDG), the Interne Fondsen KU Leuven / Internal Funds KU Leuven under grant agreement C14/18/094 (GB), and the Research Foundation—Flanders ("Fonds voor Wetenschappelijk Onderzoek—Vlaanderen," G0E1420N; GB). This work was funded by Fundação de Amparo à Ciência e Tecnologia do Estado de Pernambuco (APQ-0078-2.02/16) and Conselho Nacional de Desenvolvimento Científico e Tecnológico (CNPq) for the research Grant PQ-2 of Wallau, GL (303902/2019-1). We thank the Bioinformatics Core (NBI) and Technological Platform Core (NPT) of the Instituto Aggeu Magalhães for the computational analysis server and NGS sequencing.

## Author contributions

Designed the study: A.F.B., T.A.P., R.F.O.F., G.L.W. and N.D.G. Performed the sequencing: L.C.M., M.J.L.S., J.R.F., R.D.O.C. and C.C.K. Data curation: A.F.B., J.R.F., R.J.O., M.E.P., E.A., G.C.E., Q.M.T., A.T.H. and D.A.T.C. Analyzed the data: A.F.B., J.R.F., R.J.O., M.E.P., G.C.E., Q.M.T., G.B. and T.A.P. Provided clinical samples and/or resources: L.C.M., M.J.L.S., R.D.O.C., F.Z.D., M.R.P., L.A.C.-J., E.C.M.M., L.M.R.P., R.F.O.F., B.A.L.F. and G.L.W. Writing—original draft preparation: A.F.B., J.R.F., R.J.O., M.E.P., T.A.P. and N.D.G. Writing—review and editing: A.F.B., R.F.O.F., Q.M.T., T.A.P., G.L.W. and N.D.G. Supervision: G.B., R.F.O.F., T.A.P., B.A.L.F., G.L.W. and N.D.G.

## Competing interests

The authors declare no competing interests.
