## [Peer Review File · Nature Communications]

Reviewers' Comments:

Reviewer #2:

Remarks to the Author:

As per my previous review, I still find this study to be of high quality, yet appreciate some of the more critical comments of the other two reviewers. Several of these comments were related to novelty and the strength of conclusions which may be drawn. I believe that the authors did a good job of responding to these comments, particularly with the toning down of language and more precise qualifying of certain conclusions (as well as the additional inclusion of serological data). The authors pose several new hypotheses based on their findings which may open new avenues of research, even if their present analyses based on currently available data cannot yet fully prove these hypotheses.

Reviewer #3:

Remarks to the Author:

Overall comments:

The authors are to be commended for substantially revising and improving the text and methods of the submission. The revised text is significantly strengthened relative to the original version and is more expansive in its scope, while better highlighting caveats in the data, modeling, and interpretations. The revised text is also well written, clear, and more broadly accessible to non-flavivirus audiences, enhancing its appeal to a wide audience. As it stands, the text advances the understanding of the decline in dengue incidence that was observed in many Latin American countries following the large Zika pandemic in 2016.

Minor comments:

1. Line 231 – “higher dengue cases” should be replaced with “higher dengue case counts” or “higher dengue incidence”, as appropriate.
2. Line 393 – The PAHO acronym should be defined, as not all readers, even in the health/medical area, will be familiar with this term.
3. Line 428 – “Future studies...should ideally sample genomes following proportions and distributions matching the disease burden in times and locations under investigation.” It is not clear what a proportion/distribution of disease burden means. Do the authors mean “...sample genomes commensurate with the disease burden...”? Please clarify or simplify this sentence.
4. Figure S7 – The NTC acronym should be defined.

Reviewer #3 (Remarks to the Author):

Minor comments:

1. Line 231 – “higher dengue cases” should be replaced with “higher dengue case counts” or “higher dengue incidence”, as appropriate.

That sentence was corrected, and now reads as follows:

*“To investigate, we sequenced DENV from the recent outbreaks in Northeast Brazil (mainly affected by DENV-1) and Southeast Brazil (mainly affected by DENV-2)³¹, regions that typically have **higher dengue case counts** than the rest of the country (Figure 1C).”*

2. Line 393 – The PAHO acronym should be defined, as not all readers, even in the health/medical area, will be familiar with this term.

That sentence now reads as follows:

*“After increases in the number of Guillain-Barré Syndrome and microcephaly in northeastern Brazilian states in late 2015, the **Pan American Health Organization (PAHO) and the World Health Organization (WHO)** deployed experts of the Global Outbreak Alert and Response Network (GOARN) to Brazil in November 2015.”*

3. Line 428 – “Future studies...should ideally sample genomes following proportions and distributions matching the disease burden in times and locations under investigation.” It is not clear what a proportion/distribution of disease burden means. Do the authors mean “...sample genomes commensurate with the disease burden...”? Please clarify or simplify this sentence.

That sentence was corrected, and now reads as follows:

*“Future studies investigating the genomic epidemiology of DENV, or any pathogen, should ideally **sample genomes commensurate with the disease incidence** in times and locations under investigation (Figure 2), to ensure more realistic reconstructions of the geographic patterns of spread.”*

4. Figure S7 – The NTC acronym should be defined.

The legend of Figure S7 now includes a definition of NTC.

*“Figure S7. Genome coverage of DENV-1 and DENV-2 samples sequenced from FTA filter paper cards. Dotplot showing the percentage of the genome covered by >10X depth by the overall proportion of reads aligning to the DENV genome generated. Dashed line represents a 70% genome coverage threshold. NTC = **No template control** (“**Water control**”).”*